# An Improved Energy Management Strategy of Diesel-Electric Hybrid Propulsion System Based on FNN-DP Strategy

Wang Li [1], Chao Wang [2] , Haoying Pei [2], Chunmei Xu [2,*], Gengyi Lin [2], Jiangming Deng [1], Dafa Jiang [1] and Yuanju Huang [1]

1   State Key Laboratory of Heavy Duty AC Drive Electric Locomotive Systems Integration, CRRC Zhuzhou Locomotive Co., Ltd., Zhuzhou 412001, China
2   School of Electrical Engineering, Beijing Jiaotong University, Beijing 100044, China
*   Correspondence: chmxu@bjtu.edu.cn

**Abstract:** Diesel-electric hybrid propulsion system (HPS) is widely applied for shunting locomotive due to the characteristics of flexible configuration, economic and environmental protection in the world. Energy management strategy (EMS) is an important design factor of HPS that can optimize the energy distribution of each power sources, improve system efficiency, and reduce fuel consumption. In this paper, the model of HPS for shunting locomotive and system operating profile are firstly carried out. Then the EMS consist of the conventional rule-based (RB) strategy rule, and a fuzzy neural network base on dynamic programming (FNN-DP) strategy are studied. Finally, the simulations were carried out with these EMSs in the system model at full operating conditions to derive the fuel consumption. The conclusion is that the theoretical optimal solution of DP provides reference and guidance for the fuzzy neural network strategy to improve the rules, and the fuel consumption of the FNN-DP strategy is 10.2% lower than the conventional RB strategy.

**Keywords:** diesel-electric hybrid propulsion system; dynamic programming; energy management strategy; fuzzy neural network



## 1. Introduction

In railway transportation, the shunting locomotive is essential. It is commonly utilized in both maintenance hubs and operational routes to move rolling stock cars, supplies, and equipment. In addition, when an accident happens, the shunting locomotive is responsible for retrieving damaged or powerless vehicles in order to ensure passenger safety [1,2]. In terms of shunting locomotive market demand, relative research indicates that there will be a need for at least 2000 units in the future in China from two perspectives. The DF5, DF7, and DF4D locomotives, which were produced between the 1980s and the beginning of the twenty-first century, are waiting to be replaced [3]. Second, China's aggressive implementation of the "road-to-rail" strategy necessitates the development of the shunting locomotive market to match the rapid growth in rail freight traffic. Therefore, it is urgent to develop energy integration technology to meet the development of electric locomotive market and energy conservation and emission reduction [4]. With the development of energy storage system, hybrid propulsion system is widely to solve this problem [5].

Due to the large-scale and long-distance demands of locomotive working characteristics, diesel-electric hybrid propulsion system (HPS) is being intensively developed across the globe in response to the need for energy savings and pollution reduction [6,7]. Diesel-electric HPS is usually made up of a diesel generator and a battery pack. This system is adaptable and can handle any shunting duty. It also has a high torque even at a low running speed, allowing the diesel engine to operate in the high efficiency zone all of the time, considerably boosting fuel economy, and decreasing mechanical vibration [8,9].

Due to the different power and energy characteristics of diesel generator and battery pack, reasonable energy distribution of different power sources under specific load profile is

the most important method to ensure the stability and energy conservation of HPS [10,11]. The different output characteristics of different power sources lead to more complex energy management, which may have a negative impact on the system [12]. Energy management strategy (EMS), which integrates the locomotive's real-time operation state with the system's status information and determines the operation orders for each power source under various operating situations, is proposed to ensure the whole system operates in an optimum or near-optimal mode [13,14].

Most researches have studied EMS for various vehicles. There are two types of EMS now available: rule-based (RB) strategy, optimization-based (OB) strategy [15,16]. The RB strategy is usually separated into logic and fuzzy control rules which are based on a large number of driving experiences. These strategies are easy to regulate and use in real time, but they are not flexible to changing load profiles and cannot attain the optimum power system efficiency in reality. The OB strategy combines fuel consumption, economic cost, and dynamics to accomplish power distribution by determining the minimal objective function value, which is separated into global and instantaneous optimization approaches. Nazari, et al. [17] used dynamic programming (DP) algorithm to obtain the global optimal power allocation results. The algorithm is based on the goal of total cost of power consumption of the multi-energy storage system throughout its whole life cycle, and capacity parameters are adjusted based on the DP results. To obtain the configuration guidance program, Herrera, et al. [18] used the traditional threshold method as EMS, which is based on genetic algorithm (GA), to optimize the energy storage system over the full life cycle of the initial acquisition cost, replacement cost, operating electricity costs, and other multi-objectives. Based on these researches, the global optimization strategy can get the best energy distribution result to reduce the fuel consumption of HPS. However, the global optimization strategy requires knowing the driving conditions of the route in order to obtain the global optimal solution, but it cannot achieve real-time control effect. For this reason, relevant scholars have proposed to achieve better optimization performance by combining optimization algorithm with deep learning [19,20].

In order to meet the real-time energy management of HPS and reduce fuel consumption at the same time, this paper proposes an improved EMS which combines fuzzy neural network (FNN) and DP strategy. FNN-DP strategy solves the defect that DP strategy cannot be used for real-time vehicle operation, and provides approximate optimization effect. In this strategy, FNN is used to optimize the fuzzy rules based on DP's optimal energy distribution results. This method can realize the real-time energy management of the hybrid power system on the basis of approximately realizing the DP global optimization effect.

## 2. HPS for Shunting Locomotive

This chapter develops a locomotive HPS model for a shunting locomotive and completes the system operating profile based on the real line's load profile, establishing the groundwork for the following EMS.

### 2.1. The Model of HPS for Shunting Locomotive

The topological structure of HPS for a shunting locomotive is shown in Figure 1 [21]. The main circuit is a series structure, consisting of a diesel generator, a battery group, DC/DC converters, AC/DC rectifies, two sets of traction inverters, one set of auxiliary inverters, and four traction motor [22]. This paper mainly studies the energy distribution of the diesel generator and battery pack, so the model of drive system which combine inverters and traction motors is replaced by load profile.

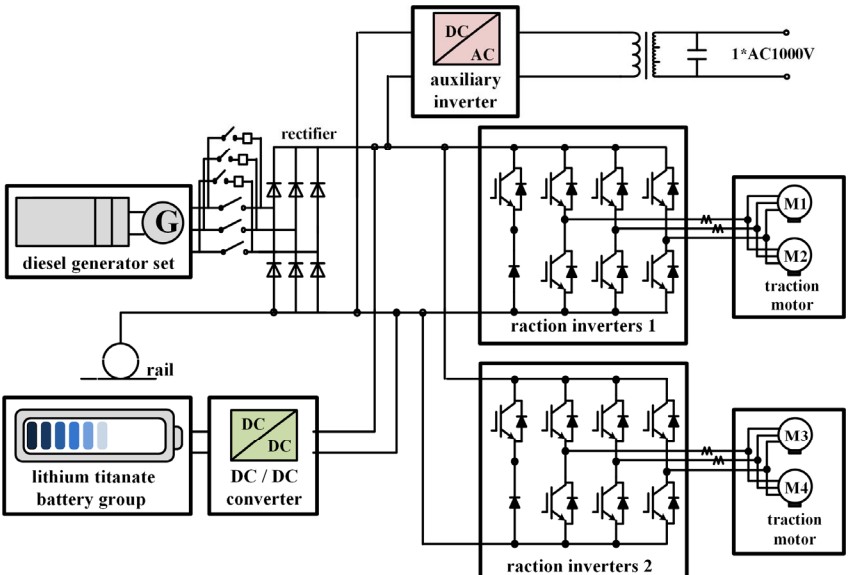

**Figure 1.** Topological structure of HPS for shunting locomotive.

*2.2. Power Sources Dynamic Energy Model*

2.2.1. Diesel Generator Dynamic Characteristics

In locomotive applications, gensets composed by prime movers (either diesel engine or gas turbine) and well-proven alternators are the most important power sources. So far, a diesel engine with a synchronous generator (SG) is the mainstream choice for Diesel generator (DG). Meanwhile, a high-speed gas turbine coupled with a permanent magnet generator (PMG) is drawing attention for its better efficiency and reduced volume. In the case, an active or passive rectifier is also mandatory to utilize the power of gensets. Due to industrial concerns of cost and robustness, passive rectifiers are currently more frequently used in present-stage. It is noteworthy that passive rectifiers cannot achieve decoupled control of output power, which means the mechanical dynamics will affect the transient of the DG. Therefore, it is necessary to model the mechanical part while analyzing DG. In this paper, mature generator models provided by *SimPowerSystem* are employed, while the mechanical part is approximately modeled by a conventional PID controller, an actuator, and engine delay as shown in Figure 2, formulated as

$$T_m(s) = \frac{K_{act}}{1+\tau_{act}s} \cdot K_{de}e^{-\tau_{de}s} \cdot Y(s)$$
$$\tau_{act} \approx 0.9/2\pi n, \tau_{act} \approx 1/2nN \tag{1}$$

where $T_m$ is the mechanical torque, $K_{act}$ is the actuator gain, $K_{de}$ is the engine torque gain, $J$ is the moment of inertia, $n$ is the rotating speed of the coaxial structure, and $N$ is the number of cylinders.

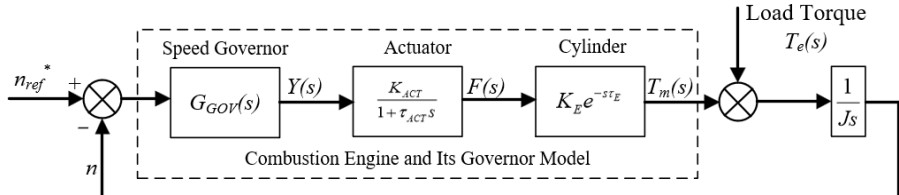

**Figure 2.** Block diagram of the reduce-order mechanical model for genset.

The average value function is given as:

$$V_{dc} = \frac{3\sqrt{3}}{\pi}V_m - \frac{3}{\pi}\omega_e L_{ac}I_{dc} \tag{2}$$

where $V_m$ is the peak value of phase voltage, $\omega_e$ is the electrical angular speed, $L_{ac}$ is the ac-side inductance (i.e., synchronous inductance of the SG), and $I_{dc}$ is the average value of output current.

### 2.2.2. Battery Dynamic Characteristics

The battery model used in this paper is an internal resistance model, which is mainly composed of a nonlinear controllable voltage source and a series resistance, as shown in Figure 3.

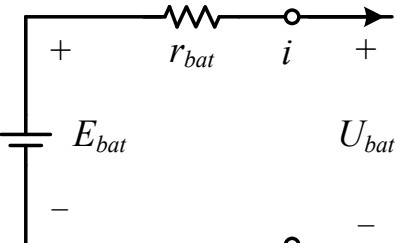

**Figure 3.** Equivalent circuit of battery.

The output voltage $U_{bat}$ of the battery is:

$$U_{bat} = E_{bat} - i \cdot r_{bat} \tag{3}$$

where, $U_{bat}$ is the output voltage of the battery pack. $E_{bat}$ is the open circuit voltage of the battery pack. $i$ is the output current of the battery pack. $r_{bat}$ is the internal resistance of the battery pack.

In the battery model, because the battery can hold a certain amount of electricity, the energy required to be charged and discharged by the battery is usually determined by calculating the state of charge (SOC) of the battery. According to the commonly used Perkert formula, the capacity of the battery pack can be expressed by Equation (4).

$$Q = i^n t \tag{4}$$

where, $Q$ is the capacity. $n$ is constant of Perkert. $t$ is the time of charging or discharging.

Therefore, as one of the key variables reflecting the state of the battery and an important input variable in the subsequent energy management system, the SOC of the battery can be calculated by (5).

$$SOC = 100 \cdot \left[ \frac{Q_o - \int_0^t i\,dt}{Q_{\max}} \right] \tag{5}$$

where, $Q_o$ is the initial value of the battery's state of charge. $Q_{\max}$ is the maximum capacity of the battery.

The open circuit voltage of the battery pack is related to the input and output current and the state of charge of the battery. It can be expressed by Equations (6) and (7) when ignoring the hysteresis effect of temperature factors on the battery. The battery pack model consists of charging model and discharging model.

When discharging, the battery output voltage is:

$$U_{bat\_dis} = E_o - K \cdot \frac{Q_{\max}}{Q_{\max} - it} \cdot i^* - K \cdot \frac{Q_{\max}}{Q_{\max} - it} \cdot it + A \cdot \exp(-B \cdot it) \tag{6}$$

When charging, the battery output voltage is:

$$U_{bat\_ch} = E_o - K \cdot \frac{Q_{\max}}{it + 0.1 \cdot Q_{\max}} \cdot i^* - K \cdot \frac{Q_{\max}}{Q_{\max} - it} \cdot it + A \cdot \exp(-B \cdot it) \tag{7}$$

where, $E_o$ is the constant voltage value of the battery pack. $K$ is the polarization constant. $i_t$ is the extracted capacity of the battery pack. $i*$ is low frequency dynamic current. $A$ is the exponential voltage. $B$ is the exponential capacity.

### 2.3. System Load Profile

The combined force diagram of traction force, braking force, and resistance at each speed moment is plotted using the traction characteristic curve and the braking characteristic curve, and the time and distance derivation formula in the speed interval is calculated [23]. After obtaining the operating circumstances (distance-speed and distance-time relationship graphs), the traction computation is programmed to produce the locomotive's load profile. The framework of traction calculation is shown in Figure 4.

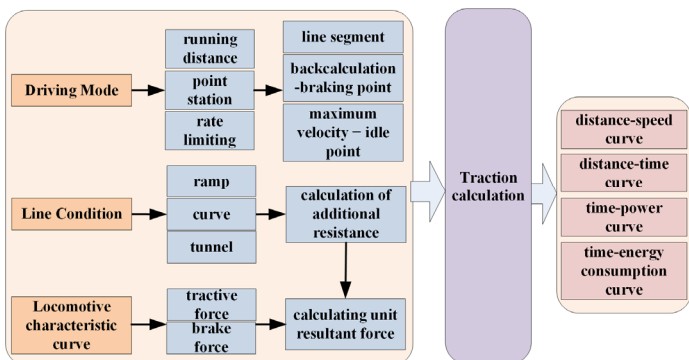

**Figure 4.** The framework of traction calculation.

### 3. Research of Energy Management Strategy

EMS is the core of system operation [24]. An efficient EMS can assure steady locomotive operation and reduce fuel consumption. Based on the model of HPS, this proposed a fuzzy neural network base on dynamic programming (FNN-DP) strategy. The proposed FNN-DP strategy combines FNN and DP strategy in order to meet minimum fuel consumption and real-time performance at the same time.

### 3.1. RB Strategies

The RB strategy does not need to predict operating circumstances and can provide real-time results. The real demand algorithm technique is separated into two cases: demand power less than 0 and demand power larger than 0. The specific flow block diagrams are shown in Figures 5 and 6 below.

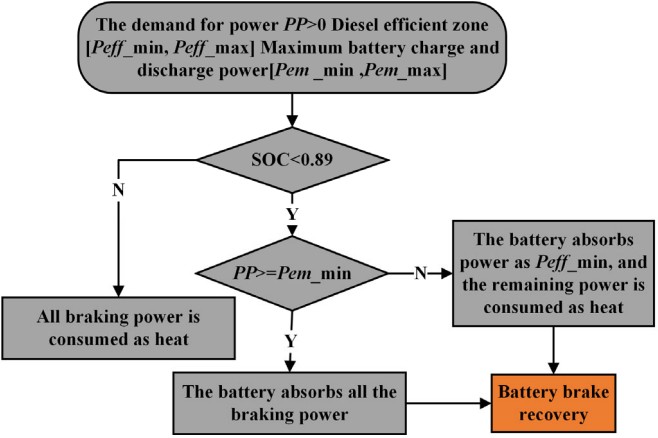

**Figure 5.** Flow chart of RB strategy when *PP* < 0.

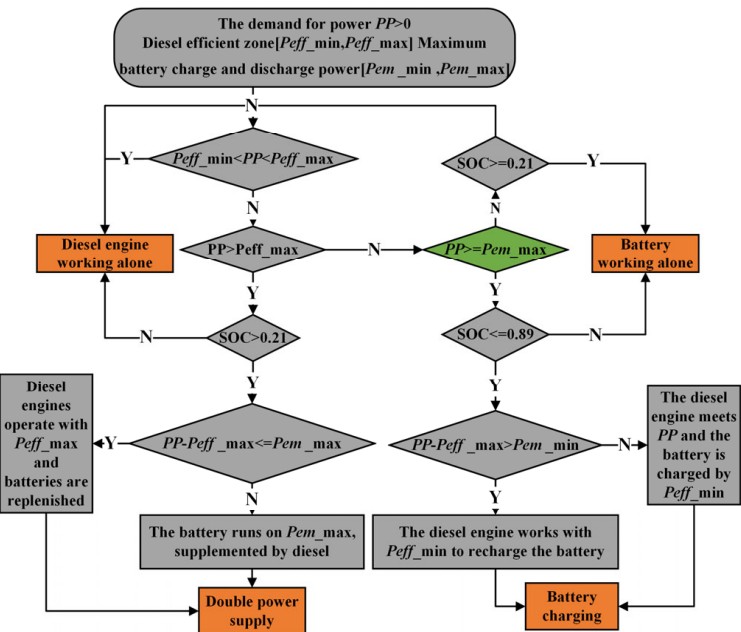

**Figure 6.** Flow chart of RB strategy when *PP* > 0.

Among them, in the high efficiency zone, the output power of a diesel generator set is [*Peff* _min, *Peff* _max]; the demand power in specific working condition is *PP*; the maximum charging and discharging power of battery is [*Pem*_min, *Pem*_max]; the state of charge (SOC) range of battery group is [0.2,0.9]. The basic rule is to use the diesel engine as the primary energy source, to work in the most efficient range possible (lower fuel consumption and emissions), and to use the battery to meet the remaining demand power as much as possible, or to absorb excess power and recover brake power as much as possible. The goal of the RB strategy is to keep the engine running in the range of high efficiency output power, which means that fuel consumption is low in the range, and the battery group to make up or absorb the remaining power. If the remaining power exceeds the top limit of the battery pack's maximum working capacity, the battery pack should first be checked for proper functioning by increasing the diesel generator's output power.

The RB strategy is simpler and faster to calculate, can be invoked in real time, and can reduce the fuel consumption and emissions of HPS to some extent. However, the rules are based primarily on the engineering experience of research scholars, resulting in imperfect rules that do not guarantee that the system achieves optimal control [25].

### 3.2. DP-Based Global Optimization Control Strategy

Richard Bellman devised and developed DP in the 1950s. Now, in a HPS for locomotives, DP strategy is a mathematical approach for designing an optimum energy management controller and managing the ideal operating state of two energy sources. The benefit of the DP strategy over other EMSs is that it can achieve the global optimum scheme, which may be utilized as a benchmark tool [26]. The global optimality criteria seeks to maximize the whole predicted path rather than the power system's operating condition at any one time. As a result, solving the energy consumption at a single point is pointless, and all energy consumption throughout the journey must be addressed. To put it another way, DP must strike a balance between present low energy usage and projected high energy demand in the future.

Figure 7 depicts the core premise of global energy management optimization for multi-energy drive systems using the DP algorithm.

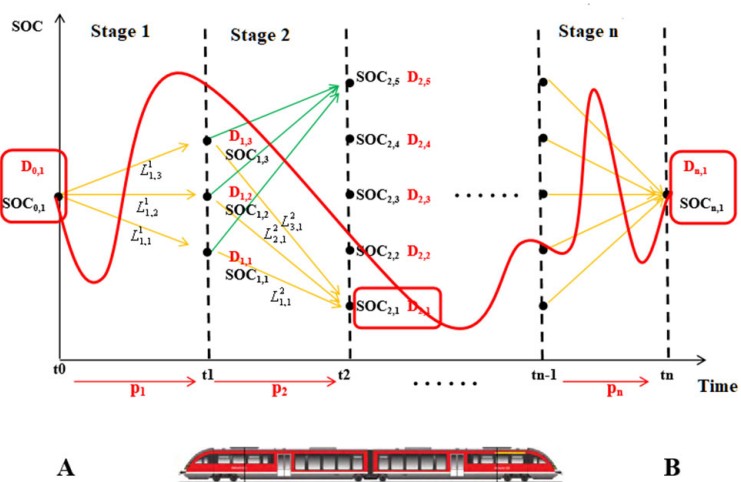

**Figure 7.** DP basic principle diagram.

The purpose of DP is to obtain the best possible fuel efficiency for the HPS under all circumstances. In the diagram, the time of the abscissa $t_0 \sim t_n$, when the train runs from point A to point B, is discretized into n stages, and the ordinate is discretized into m state variables *SOC*, which represent the battery's remaining power. The algorithm's ultimate purpose is to determine the best *SOC* trajectory to fulfill the least fuel consumption constraints. This trajectory's changing trend varies depending on the output of the diesel generator and the battery pack. $SOC_{0,1}$ is the initial state of charge of the battery pack, $D_{i,j}$ is the total fuel consumption accumulated from the initial point to the state point *j* in the stage *i*, which is a one-step cost function, that is, the fuel consumption between the state point $j_2$ in the *i* stage and the state point $j_2$ in the previous stage.

1. Record the fuel consumption of each state point from the initial point $SOC_{0,1}$ to stage 1, that is *L*, the corresponding cumulative fuel consumption;
2. Calculate the *L* of each state point from stage 1 to stage 2, and only keep track of the total minimum fuel consumption from the beginning to each stage 2 state point. As shown in point $SOC_{2,1}$ in the figure, select the route with the lowest cumulative fuel consumption, that is, one of the three yellow lines, and store the relevant data of this route to the corresponding matrix to facilitate backtracking;
3. Step 2 should be repeated iteratively until the state is ended. The termination phase is the same as the beginning phase, with the goal of ensuring that all cumulative fuel consumption is from diesel engines without the use of electric energy. Iterative cumulation is used to determine the transfer cost of single step state variables, and a set of control variables with the lowest cumulative cost in the whole discrete space is produced. The DP optimum solution is this set of control sequences. State variable, decision variable, state transfer equation, constraint condition, single stage objective function, and total objective function must all be calculated throughout the procedure.

(a) State variable

The state variable in this study is the battery's *SOC*, and the stage division step size is 0.0001. The state variable *x* and its step size are shown in Equation (8):

$$x(k) = SOC(k)$$
$$\Delta x = 0.0001 \tag{8}$$

(b) Decision variables

The output power of the battery pack ($P_{batt}$) is chosen as the decision variable in this paper, and each stage's decision forms a set of control variable sequence. The stage step TS is 1s, and $E_{batt}$ is the battery power. The decision variable u and its step size are shown in Equation (9):

$$\begin{aligned} u(k) &= P_{batt}(k) \\ \Delta u &= \frac{3600 \times E_{batt} \times \Delta x}{T_s} \end{aligned} \tag{9}$$

(c) State transfer equation

SOC changes in the whole operating situation at any moment due to diverse decision-making options. The state is passed from one stage to the next in the DP iterative process via making choices, as shown in Equation (10):

$$\begin{aligned} SOC^j(k+1) &= SOC^i(k) + \frac{3600 \times P_{batt}^{ij} \times T_s}{E_{batt}} \\ k &\in \{1,2,\ldots,n\}, i,j \in \{1,2,\ldots,m\} \end{aligned} \tag{10}$$

(d) Constraint condition

The constraints to be satisfied in the DP iteration process are shown in the following Equations (11) and in (12):

$$\begin{aligned} P_{req} &= P_{em} + P_{DG} \\ P_{DG} &= P_{ice} \times \eta_{ice} \\ P_{em} &= P_{batt} \times \eta_{DC/DC} \end{aligned} \tag{11}$$

$$\begin{aligned} SOC_{min} &\leq SOC \leq SOC_{max} \\ P_{ice\_min} &\leq P_{ice} \leq P_{ice\_max} \\ P_{req\_min} &\leq P_{batt} \leq P_{req\_max} \\ P_{batt\_min} &\leq P_{batt} \leq P_{batt\_max} \end{aligned} \tag{12}$$

The aim function at the single step stage of DP is low fuel consumption, which includes not only the diesel directly used by the engine, but also the fuel indirectly absorbed by the battery. As shown in Equations (13) and (14). In Equation (15) is the fuel consumption curve of the diesel generator, which is a function of the output power of the diesel generator.

$$\begin{aligned} L(k) &= FC(k) = FC_{DG}(x(k), u(k)) + FC_{batt}(x(k), u(k)) \\ FC_{DG}(k) &= \frac{P_{ice}(k) \times Con_{DG}(k)}{3600 \times 1000} \end{aligned} \tag{13}$$

$$\begin{aligned} Con_{DG}(k) &= 1.389\, P_{ice}(k), P_{ice} \subset [0, 180] \\ Con_{DG}(k) &= 0.000132 P_{ice}^2(k) - 0.17 P_{ice}(k) + 230.18, P_{ice} \subset [180, P_{ice\_max}] \end{aligned} \tag{14}$$

$$FC_{batt}(k) = \frac{E_{batt}}{H_l \times s \times \eta_{ice}} \tag{15}$$

(e) Single stage objective function

The whole cycle condition is divided into n stages, and the value of the goal function for each of the N single-step stages is added together. The following Equation (16) expresses the overall objective function of the DP energy management technique in terms of state and control variables.

$$D = min \sum_{k=1}^{n} L(k)\Delta t = min \sum_{k=1}^{n} FC(k)T_s \tag{16}$$

Among them, the meaning of the relevant symbols is shown in the Table 1, and the calculation flow chart of DP strategy is shown in Figure 8.

**Table 1.** Nomenclature.

| Symbol | Meaning |
| --- | --- |
| $E_{batt}$ | The Battery pack capacity |
| $P_{batt}^{i,j}$ | Decision variables from state $i$ to state $j$ |
| $P_{req}$ | DC side demand power |
| $P_{em}$, $P_{batt}$ | Generator and battery power |
| $P_{DG}$, $P_{ice}$ | Diesel generator set and engine power |
| $\eta_{ice}$, $\eta_{DC/DC}$ | Efficiency of diesel engine and DC/DC converter |
| $FC_{batt}$ | Fuel consumption of diesel engine set |
| $FC_{DG}$ | Equivalent fuel consumption of battery pack |
| $Con_{DG}$ | Fitting curve of diesel consumption |
| $H_l$ | Calorific value of diesel oil |
| $s$ | Oil electricity conversion factor |

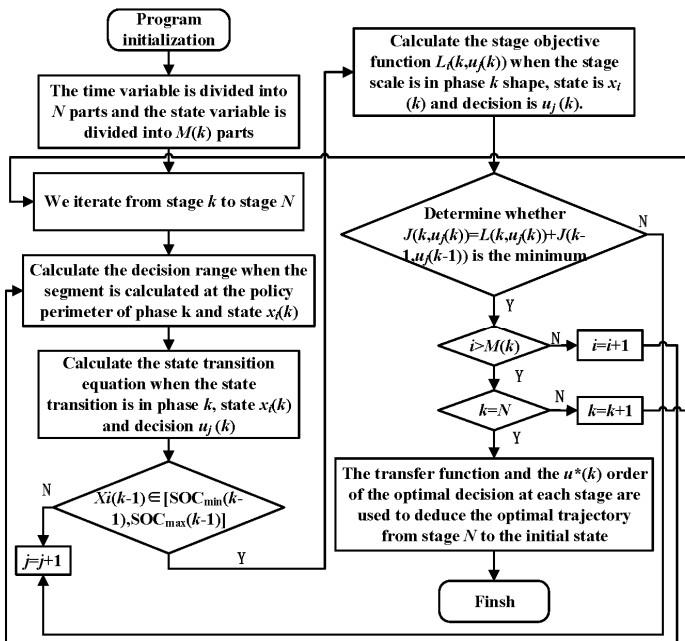

**Figure 8.** DP strategy flow block diagram.

### 3.3. FNN-DP Strategy

Although the global optimal EMS (DP strategy) can maximize the performance of the HPS to achieve the global optimum [22], thus improving the vehicle's fuel economy, it can only be used to obtain information on future driving conditions in order to perform optimization calculations, while not avoiding the algorithm's shortcomings of large computational volume and long computation time, and thus belongs to the offline algorithm, which is limited in real-time applications. In this paper, a fuzzy neural network (FNN) is proposed to optimize the fuzzy rules based on the DP strategy's global optimized energy distribution results. At each moment of locomotive real-time operation, real-time energy management can be carried out through fuzzy control rules determined by FNN-DP strategy according to load power and battery *SOC*.

The typical structure of fuzzy neural network is shown in Figure 9. There are four layers in the structure. According to the function, it is divided into input layer, fuzzification layer, fuzzy reasoning layer and anti-fuzzy layer. The membership function of fuzzification layer and the weight between layers can be adjusted. It can be seen that each layer and each node of the neural network can be combined with fuzzy rules, so that the adjustment of fuzzy rules can no longer rely on subjective experience, and can give clear physical meaning to various parameters in the network structure.

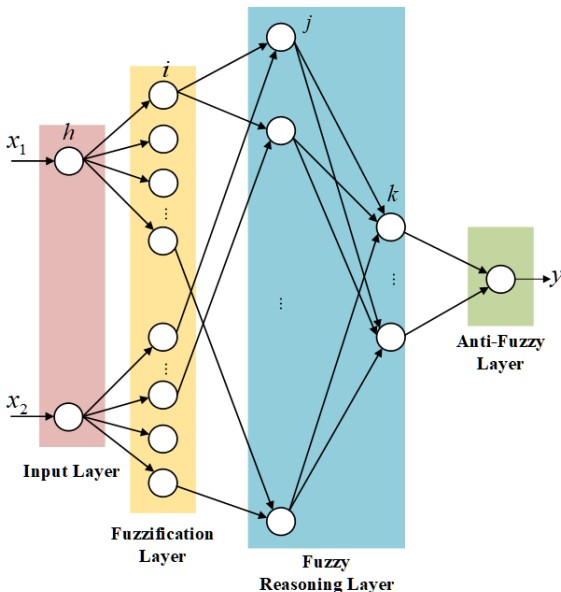

**Figure 9.** Typical structure of fuzzy neural network.

### 3.3.1. Fuzzy Rules Representation

Firstly, the input of fuzzy neural network controller is load demand power ($P_{load}$) and battery's *SOC*, and the output is diesel generator set power ($P_G$). By calculating the difference between the load demand power and the power of the diesel generator set, the output power of the power battery pack is derived. Because the diesel generator does not output power when absorbing regenerative braking energy, the load demand power is less than zero. In the design of fuzzy control strategy, the load demand power is divided into five grades from 0 to 950 kW according to the size, and the fuzzy domain is converted to [0,1]. The range of 20% to 80% of *SOC* can meet the needs of power battery pack operation, and its domain is defined as [0.2,0.8]. Similarly, the power of diesel generator set is converted to [0,1]. The above rules are transformed into fuzzy logic rules as Table 2.

**Table 2.** Fuzzy logic rule table.

| $P_G$ | | *SOC* | | |
|---|---|---|---|---|
| | | **S** | **M** | **B** |
| | VS | S | VS | VS |
| | S | M | S | S |
| $P_{load}$ | M | B | M | M |
| | B | B | B | B |
| | VB | VB | B | B |

Selecting gauss function as membership function of fuzzy rules, as shown in Equation (17):

$$\mu_A(x) = \exp\left(-\left(\frac{x-a}{b}\right)^2\right) \qquad (17)$$

where $a$ is the central value of gaussian function; $b$ is the width value of gaussian function.

The fuzzy neural network is accurately determined according to the results of the selected membership function and rule matching calculation. Therefore, the center value and width value of the Gaussian function are very important for the fuzzy processing of the fuzzy neural network calculation process and the final output result. In the case that the output result cannot reach the expectation, it is necessary to use the learning function of the neural network to set.

### 3.3.2. Fuzzy Neural Network Controller Structure

In the fuzzy neural network controller, the load demand power ($P_{load}$) and the *SOC* are input variables, and the output variable is the diesel generator output power. In this paper, a fuzzy neural network based on Takagi-Sugeno (TS) model is used. As shown in Figure 10, T-S fuzzy inference has the advantages of simple calculation and good for mathematical analysis. It is easy to realize the adaptive function of the controller. The controller is composed of antecedent network and consequent network. The function of antecedent network is to match rules and calculate matching weight. The consequent network is a fuzzy rule processing of input variables.

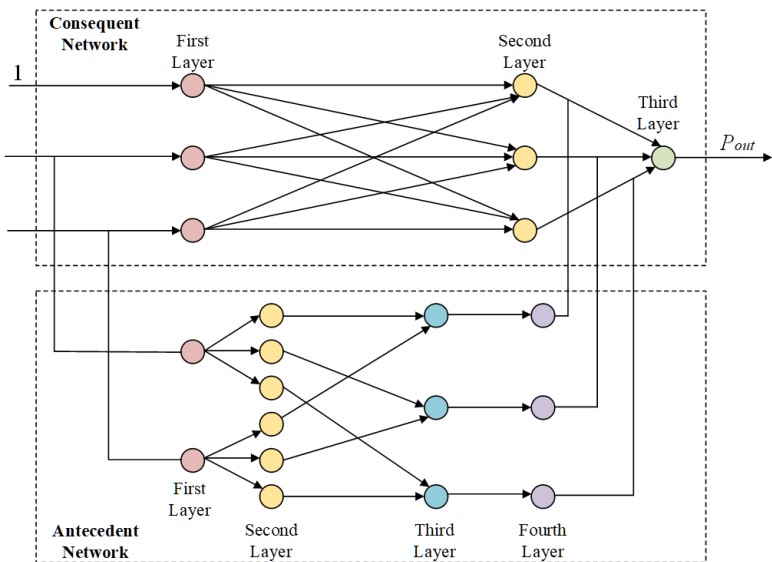

**Figure 10.** Fuzzy neural network controller structure.

(a)    Antecedent network

The antecedent network has four layers [27]. The first layer is the input layer, and the two input variables are the exact values, namely the load demand power ($P_{load}$) and the battery *SOC.* After entering the controller structure, it enters the next layer for fuzzification, and the system input has a total of 2 neurons, as shown in Equation (18).

$$x = [x_1, x_2]^T \qquad (18)$$

Membership function layer of input variables is located in the second layer, under the effect of fuzzy processing, the input is converted into fuzzy quantity. The battery SOC and load demand power language variables are divided according to the content of the previous section. The fuzzy subset of the power battery pack is {S, M, B}, and the demand power is {VS, S, M, B, VB}. Each node is the language variable defined in the fuzzy rules, so there are 8 nodes.

$$\mu_i^j = \mu A_i^j(x_i) \qquad (19)$$

where *i* is the number of input variables; *j* is the number of fuzzy partitions.

Fuzzy rules are represented by different nodes in the third layer. The purpose is to match the antecedents of fuzzy rules and calculate the respective applicability of all rules, which is expressed as the product of the output of the previous layer, as shown in Equation (20).

$$\alpha_j = \mu_1^{i1} \mu_2^{i2} \qquad (20)$$

where $i_1 = 1,2,3$; $i_2 = 1,2,3,4,5$; $j = 1,2, \dots ,15$.

The fourth layer is mainly to realize the normalized calculation, the formula is:

$$\overline{\alpha}_j = \frac{\alpha_j}{\sum_{i=1}^{m} \alpha_i (j = 1, 2, \ldots, 15)} \tag{21}$$

(b)　Consequent network

The first layer is the input layer, passing input to the next layer. There are 3 neurons in this layer. The first two are the input of the controller, and the other node is a constant value of 1, which aims to prevent the neural network from slow convergence or poor accuracy.

The main task of the second layer is to calculate the consequents of the rules. Each rule can be represented by nodes:

$$y_{ij} = p_{j0} + p_{j1} \times x_1 + p_{j2} \times x_2 \tag{22}$$

The third layer is the output layer. The role of this layer is to calculate the output of the system, the output only generator output power (PG), so the number of neurons in the third layer is only 1. In the operation of the controller, the weighted summation method is mainly used to output fuzzy consequents. The output of the antecedent network is the weighted coefficient, and the expression is:

$$y = \sum_{j=1}^{15} \overline{\alpha}_j \times y_j \tag{23}$$

### 3.3.3. Fuzzy Neural Network Learning Algorithm

In order to achieve the purpose that the error between the output result and the expected result is lower than the expectation, the weight coefficient of the fuzzy neural network needs to be optimized and adjusted, that is, the learning algorithm is used to feedback the neural network according to the error result [28]. The controller uses three parameters for learning adjustment. The three parameters are the width value ($\sigma_{ij}$) and the center value ($c_{ij}$) of the membership function ($i = 1,2$; $j = 1,2,3$), and the connection weight of the second layer of the predecessor network ($p_{ij}$) ($i = 1,2$; $j = 1,2, \ldots ,6$). When adjusting the weight coefficient, error back propagation is mainly used. In this way, learning can be more efficient. The formula for calculating the error function is as follows:

$$E = 0.5 \times (t - y)^2 \tag{24}$$

where t is the target value of the output variable; y is the real value of the output variable.

(a)　Learning algorithm of consequent network weights $p_{ij}$:

$$\frac{\partial E}{\partial p_{ji}} = \frac{\partial E}{\partial y} \times \frac{\partial y}{\partial y_j} \times \frac{\partial y_j}{\partial p_{ji}} = -\overline{\alpha}_j \times x_i \times (t - y) \tag{25}$$

$$p_{ji}(k+1) = p_{ji}(k) - \eta \times \frac{\partial E}{\partial p_{ji}} = p_{ji}(k) + \eta \times \alpha_j \times x_i \times (t - y) \tag{26}$$

where $i = 1,2$; $j = 1,2, \ldots ,15$; $\eta$ is the learning rate, $\eta > 0$.

(b)　Learning algorithm of center value ($c_{ij}$) and width value ($\sigma_{ij}$) of antecedent network

In the process of studying the antecedent network weight learning, the parameters $p_{ij}$ can be fixed, and the whole controller structure can be simplified as shown in Figure 11. In the simplified structure, the connection weights of the last layer are usually used to represent different rules in the consequent network, namely $y_{ij} = w_{ij}$.

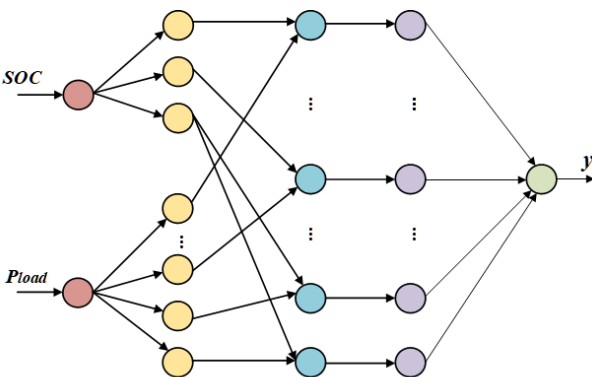

**Figure 11.** Simplified fuzzy neural network controller.

The work of fuzzy neural networks also has forward and backward propagation. The input information is first processed by forward propagation, from the input layer to the hidden layer and to the output layer. After that, through the error between the output calculation and the expected value, if the error is not within the expected range, the back propagation is carried out, and the error signal is transmitted from the output layer to the input layer. The learning algorithm is used to adjust the function structure or connection weight to achieve the purpose of reducing the error.

## 4. Improved Energy Management Strategy of Logic Threshold Rule Based on DP

Since the DP calculation is too vast and the calculation time is too lengthy, this paper takes a 15 km portion of the overall operating state to verify efforts of the proposed EMS, which encompasses the whole cycle of traction, idling, and braking [24]. Figure 12 depicts the optimum energy distribution results obtained by the DP strategy. For comparison, the results based on a conventional RB strategy are shown in Figure 13. The DP strategy needs known operating conditions ahead of time and cannot provide real-time control, but the theoretical optimum results may be used as a benchmark for improving rules of RB strategy. The difference of the diesel generator power comparison under the two strategies is shown in Figure 14. From this figure, the efficiency of diesel generator power under DP strategy is better than RB strategy. The fuel consumption comparison under two strategies is shown in Figure 15. It can be seen that DP strategy can reduce 1.0564 kg fuel consumption compared with RB strategy.

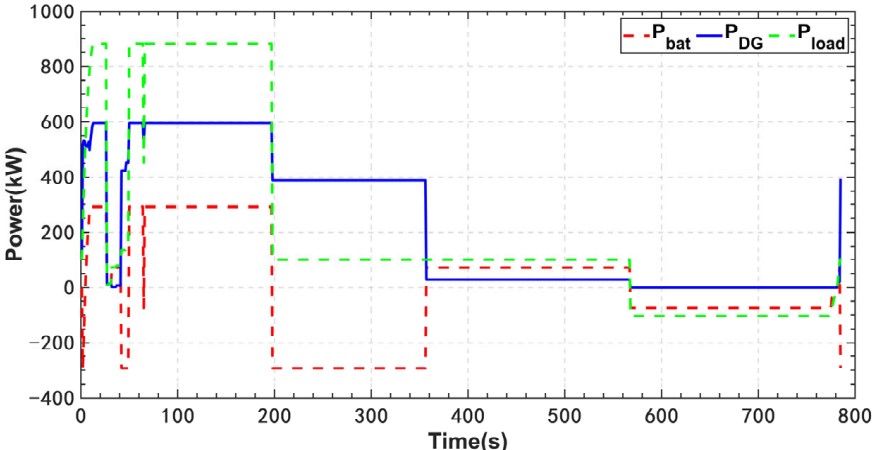

**Figure 12.** Power distribution diagram based on DP strategy.

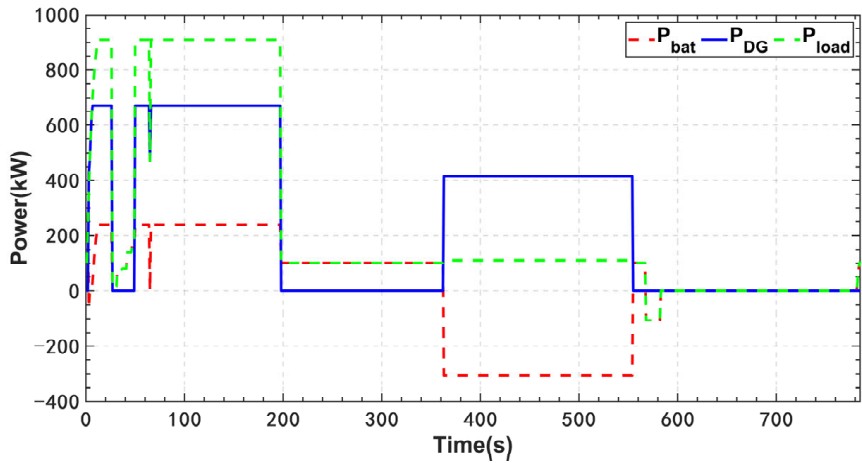

**Figure 13.** Power distribution diagram based on RB strategy.

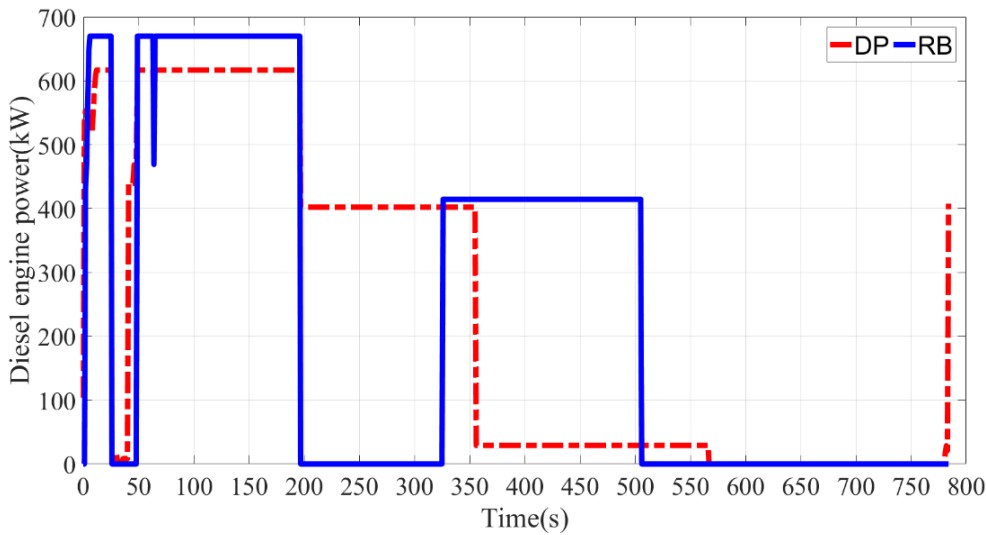

**Figure 14.** Comparison of diesel engine power between DP strategy and RB strategy.

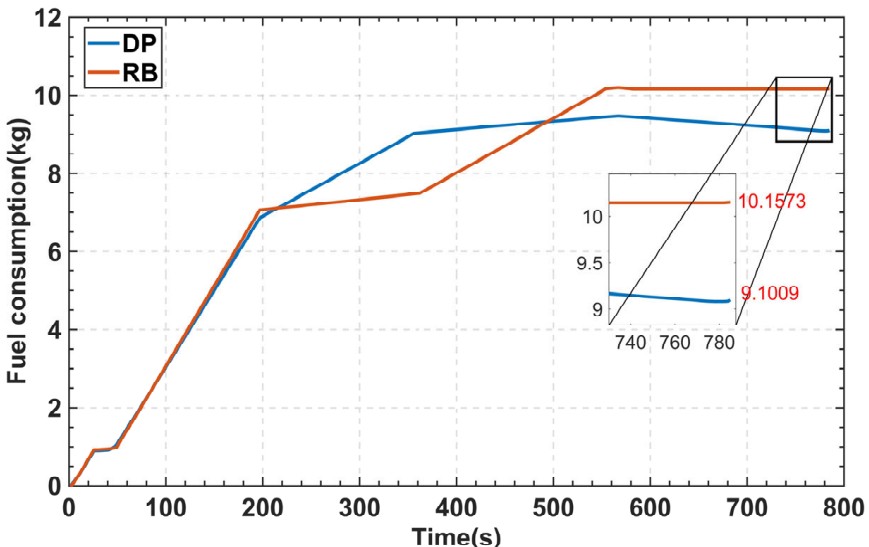

**Figure 15.** Comparison of fuel consumption between DP and logic threshold rule.

In this paper, the DP results are used as the training samples of the fuzzy neural network. The DP strategy is used to obtain the optimal allocation results of the locomotive running 784 s conditions to train the fuzzy neural network. The number of training iterations is set to 800, and 50 samples are selected as verification samples. From Figure 16, it can be seen that when the number of training iterations reaches 200, the training error curve has converged, and the error does not exceed 5%. This shows that the fuzzy rules improved by FNN-DP strategy can approximately achieve the global optimal effect of DP strategy.

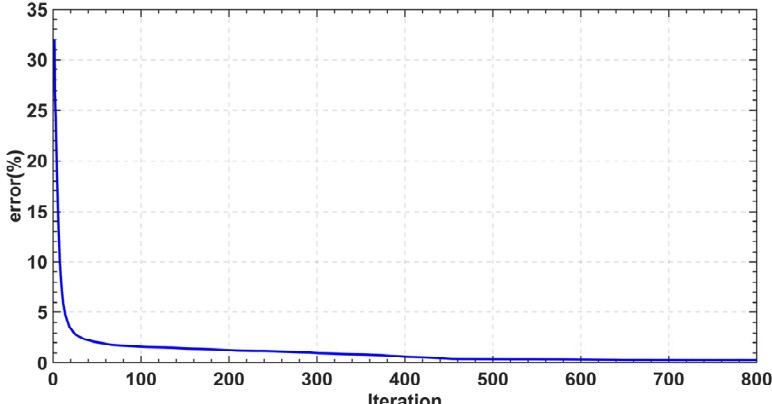

**Figure 16.** Fuzzy neural network training error convergence curve.

As shown in Figure 17, the fuel consumption of FNN-DP is 9.1210 kg, which is about 10.2% less than the RB strategy and similar to DP strategy. It shows that FNN-DP strategy has the fuel saving ability similar to the DP strategy, and can be used in the vehicle real-time energy management instead of the DP strategy.

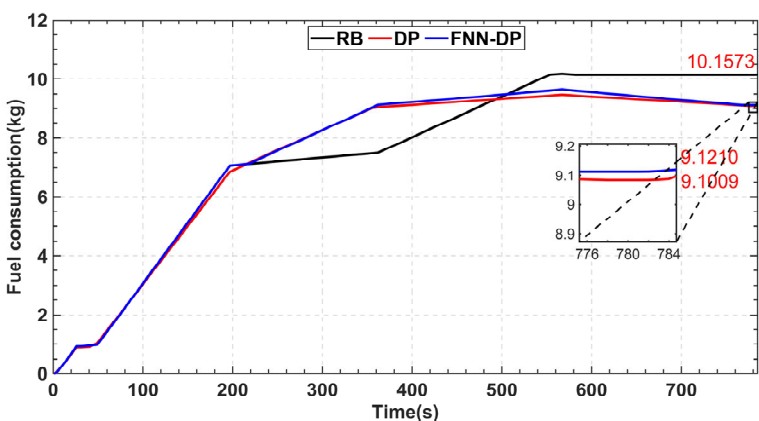

**Figure 17.** Comparison of fuel consumption between RB strategy, DP strategy and FNN-DP.

Figure 18 is the SOC comparison results under three strategies. It can be seen from the figure that the depth of discharge (DOD) of battery based on RB strategy is larger than that of DP strategy and FNN-DP strategy, which will have a negative impact on the service life of battery. The fuzzy control rules trained by FNN-DP strategy based on the energy management results optimized by DP strategy can effectively reduce the DOD of the battery, which is conducive to the healthy work of the battery. In addition, the change of battery SOC under FNN-DP strategy is similar to that of DP strategy, which can also prove that FNN-DP strategy has energy optimization effect similar to that of DP strategy.

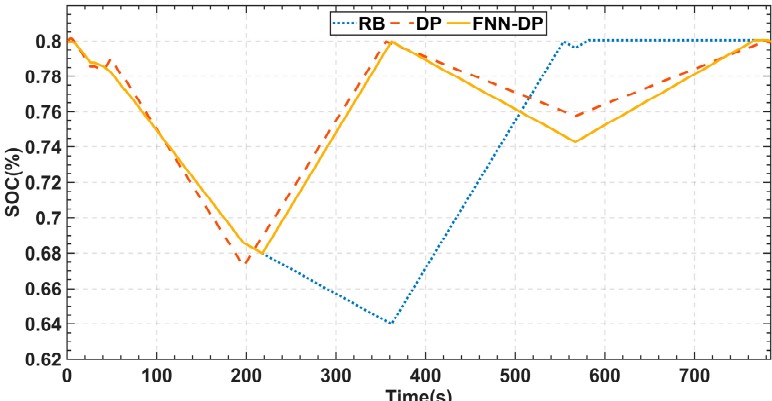

**Figure 18.** Comparison of fuel consumption between RB strategy, DP strategy and FNN-DP strategy.

## 5. Conclusions

The core of the energy management strategy for a shunting locomotive's diesel-electric hybrid propulsion system is to ensure that the responds to the desired demand of the load profile, and the process of energy conversion and transmission is optimized for control on this basis, in order to achieve a reasonable distribution of the power output of each power source system and improve the fuel economy of the entire vehicle, while maximizing the performance of the locomotive.

The advantage of RB strategy is that it is simple to calculate and can be called in real time, but the disadvantage is that the rules are not perfect and cannot guarantee that the system achieves optimal control; the advantage of DP is that it can achieve maximum optimization results, but the disadvantage is that it is computationally intensive, time consuming, and cannot be used offline in real vehicles. Therefore, this paper proposes a FNN-DP strategy, which can optimize fuzzy rules based on the global optimization energy distribution results of DP strategy, in order to meet the real-time optimization and reduce fuel consumption at the same time.

In this paper, the RB strategy and DP strategy are compared and analyzed. Finally, the theoretical optimal solution of DP is used to provide reference and guidance for the FNN-DP strategy, and can achieve improvements in both real-time and fuel savings. The findings reveal that the FNN-DP strategy reduce 10.2% fuel consumption compared with RB strategy, demonstrating the fuzzy neural network's usefulness. The fuel saving effect of FNN-DP strategy is similar to that of DP strategy, realizing the formulation of real-time rules that approximate global optimization. At the same time, according to the SOC comparison results under the three strategies, the FNN-DP strategy can effectively reduce the DOD of the battery, which is conducive to the healthy work of the battery.

Based on the research work in this paper, the proposed energy management strategy can be applied to 100% green energy electric locomotives in the future. By using new energy such as hydrogen fuel cells and batteries to form a hybrid power system, pollution-free electric locomotive drive is realized.

**Author Contributions:** Conceptualization, W.L. and C.W.; methodology, H.P.; software, H.P.; validation, J.D., D.J. and Y.H.; formal analysis, G.L.; investigation, H.P.; resources, C.X.; data curation, C.W.; writing—original draft preparation, H.P.; writing—review and editing, C.W.; supervision, C.X.; project administration, W.L. All authors have read and agreed to the published version of the manuscript.

**Funding:** This research received no external funding.

**Data Availability Statement:** Not applicable.

**Conflicts of Interest:** The authors declare no conflict of interest.

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
