# Peer review of "An Improved Energy Management Strategy of Diesel-Electric Hybrid Propulsion System Based on FNN-DP Strategy"

_electronics, doi:10.3390/electronics12030486_

Round 1

Reviewer 1 Report

1. The contributions of this paper are ambiguous, and it make reviewer puzzled. Therefore, this part should be reorganized and focused only on what is original in the paper.

2. The simulation results seems not reliable, and more validations should be provided in the paper.

3. The quality of these figures should be improved.

4. The FNN is widely implemented now. After going through the manuscript, I think the paper quality of this work is not enough. I must therefore reject it. Main drawback of this work is lack of innovation, I don't think this article takes an innovative approach.

Author Response

Response letter has shown in the following Word.

Reviewer 2 Report

1. Some other DP-based EMS strategies can be included in the literature review. The advantages of the FNN-DP based strategy over these other strategies can be specified.

2. Figure 15 illustrates the comparison of FNN-DP based strategy with the theoretical DP-based strategy. This figure can include the RB-based strategy curve as well for the best comparative analysis. 

3. Following are some minor edits:

(a) Line 53 states three types of EMS but only mentions two strategies (rule-based and optimization-based). Which is the third strategy? Or are there only two?

(b) Line 358, 359 and line 360,361 are repeated lines.

4. Suggestion for future scope: The illustration of the FNN-DP based strategy  for a complete battery-electric locomotive power system to successfully transition to a 100% green alternative.

Author Response

(The authors gave the same response as above.)

Reviewer 3 Report

-        The authors should clarify the operating conditions of each source.

-        The optimal path planning method based on real-time road conditions must be explained and analyzed.

-        A dynamic energy model must be established for this study.

-        In the EMS strategy two important elements must be discussed, these are SOC and DOD.

-        What is the originality of this work compared to other research works in the literature?

-        This proposed algorithm cannot work in real time so how the energy consumption is studied before the operation.

-        The speed and accuracy of this method should be discussed and analyzed in relation to other methods.

-        The references used are poor and this part must be improved.

-        The authors can use the articles below to improve their paper.

-            A cyber physical energy system design (CPESD) for electric vehicle applications, IEEE-SOSE2017

-            Hybrid railway traction power supply system, IEEE-IEPS 2020

-            A Fault Diagnosis Design Based on Deep Learning Approach for Electric Vehicle Applications, MDPI-Energies 2021

-           Power Conversion Technologies for a Hybrid Energy Storage System in Diesel-Electric Locomotives, IEEE Transactions on Industrial Electronics 2021

Author Response

(The authors gave the same response as above.)

Reviewer 4 Report

I believe that the subject matter of the paper is of interest, but the paper has some shortcomings that should be corrected before publication. Below are some recommendations to the authors to achieve such improvement.

1.      The use of English must be improved. The paper includes several grammatical and syntax errors.

2.      Place keywords in alphabetical order.

3.      The Introduction must be revised. Authors must go into greater detail on and analysis of the currently cited references and include more related references, like the ones listed below:

·         Sijakovic N. et al. Active System Management Approach for Flexibility Services to the Greek Transmission and Distribution System. Energies 2022, 15, 6134. https://doi.org/10.3390/en15176134

·         Fotis G., Vita V., Ekonomou L. Machine Learning Techniques for the Prediction of the Magnetic and Electric Field of Electrostatic Discharges. Electronics, Vol. 11, No. 12, 2022. https://doi.org/10.3390/electronics11121858

4.      In line 335 the authors mention: The number of training iterations is set to 800, and 50 samples are selected as verification samples Is this data set sufficient? Please refer the related literature.

5.      Please improve the resolution in the figures of your manuscript.

6.      Conclusions are not the expected according to the results obtained in the paper.

I encourage the authors to improve their work for publication.

Author Response

(The authors gave the same response as above.)

Reviewer 5 Report

This paper proposed an energy management for diesel-electric hybrid propulsion system. The technological is good. The detailed comments are as follows.

1)      In the contribution part, lines 72-76, the authors should claim the scientific contributions and advantages in details.

2)      The literature review is now sufficient. Many advanced energy management strategies are not involved, such as 1) “Event-triggered based distributed cooperative energy management for multienergy systems,” IEEE Trans. Ind. Inf., vol. 15, no. 14, pp. 2008-2022, 2019; 2) “A Survey on Distributed Optimization”, Annual Reviews in Control, vol. 47, pp. 278-305, 2019; 3) "Manage Real-Time Power Imbalance with Renewable Energy: Fast Generation Dispatch or Adaptive Frequency Regulation?,"  IEEE Transactions on Power Systems, doi: 10.1109/TPWRS.2022.3232759.

3)      A proofreading is needed for this paper.

Author Response

(The authors gave the same response as above.)

Reviewer 6 Report

The authors propose an improved energy management strategy of Diesel-Electric  hybrid propulsion system, which is the combination of a fuzzy neural network (FNN) strategy and the dynamic programming (DP) strategy.  This strategy can optimize the fuzzy rules based on the DP strategy's optimal energy distribution results. They show that the FNN-DP strategy can reduce 7.8% fuel consumption compared to the rule-based (RB) strategy. They compare the RB strategy with the DP strategy, and point out their advantage and disadvantage.

The manuscript is clearly written. The analysis is convincing. I recommend for publication in electronics.  But there is one thing I would like the authors to address.

In the conclusions, in the second paragraph, the authors state ``the advantage of DP is that it can achieve maximum optimization results, but the disadvantage is that it is computationally intensive, time consuming'' twice. Thus one of them should be deleted.

Author Response

(The authors gave the same response as above.)

Round 2

Reviewer 1 Report

Thanks for your revision. And, I have no comments.

Reviewer 3 Report

Thank you, the authors answered all my questions.

Reviewer 4 Report

The authors have revised their manuscript according to my comments. In my opinion the paper can be published in its current form.

Reviewer 5 Report

No further comments.